# Contribution of Root Hair Development to Sulfate Uptake in *Arabidopsis*

**DOI:** 10.3390/plants8040106

**Published:** 2019-04-19

**Authors:** Yuki Kimura, Tsukasa Ushiwatari, Akiko Suyama, Rumi Tominaga-Wada, Takuji Wada, Akiko Maruyama-Nakashita

**Affiliations:** 1Department of Bioscience and Biotechnology, Faculty of Agriculture, Kyushu University, 744 Motooka, Nishi-ku, Fukuoka 819-0395, Japan; YukiKimura@gmail.com (Y.K.); 2BE18410P@s.kyushu-u.ac.jp (T.U.); aksuyama@nm.beppu-u.ac.jp (A.S.); 2Graduate School of Biosphere Sciences, Hiroshima University, 1-4-4 Kagamiyama, Higashi Hiroshima, Hiroshima 739-8528, Japan; rtomi@hiroshima-u.ac.jp (R.T.-W.); twada86@hiroshima-u.ac.jp (T.W.)

**Keywords:** *Arabidopsis thaliana*, root hair, sulfate transporter, sulfate uptake, CPC, CAPRICE, WER, WERWOLF

## Abstract

Root hairs often contribute to nutrient uptake from environments, but the contribution varies among nutrients. In *Arabidopsis*, two high-affinity sulfate transporters, SULTR1;1 and SULTR1;2, are responsible for sulfate uptake by roots. Their increased expression under sulfur deficiency (−S) stimulates sulfate uptake. Inspired by the higher and lower expression, respectively, of *SULTR1;1* in mutants with more (*werwolf* [*wer*]) and fewer (*caprice* [*cpc*]) root hairs, we examined the contribution of root hairs to sulfate uptake. Sulfate uptake rates were similar among plant lines under both sulfur sufficiency (+S) and −S. Under −S, the expression of *SULTR1;1* and *SULTR1;2* was negatively correlated with the number of root hairs. These results suggest that both −S-induced *SULTR* expression and sulfate uptake rates were independent of the number of root hairs. In addition, we observed (1) a negative correlation between primary root lengths and number of root hairs and (2) a greater number of root hairs under −S than under +S. These observations suggested that under both +S and −S, sulfate uptake was influenced by the root biomass rather than the number of root hairs.

## 1. Introduction

Plant nutrients are generally absorbed from the roots in appropriate chemical forms. From the root surface to cortex, nutrients are transported via the symplasmic or apoplastic routes, but have to take the symplasmic route when they have reached the endodermis. At the endodermis, nutrients are subsequently transported to the xylem parenchyma cells and loaded into the xylem stream for further transport to shoot tissues [1,2]. The root surface cell layers, epidermis, and cortex all play important roles in nutrient uptake by expressing high-affinity transporters for most essential nutrients [2,3,4,5]. It has been suggested that increasing the root surface area exposed to the soil environments enables root hairs to also contribute to nutrient uptake [6,7,8]. Root hairs have been shown to increase the uptake of some nutrients, including phosphate and ammonium [8,9,10], but not others, such as iron and silicon [11,12]. Root hair growth is stimulated by the deficiency of phosphorous, manganese, iron, and some other nutrients in *Arabidopsis*, whereas deficiencies of other nutrients fail to produce a similar phenotype [3,9,13,14,15,16,17]. Thus, the importance of root hairs in nutrient uptake varies among nutrients.

Plants absorb sulfate as the main source of sulfur through the activity of sulfate transporters (SULTRs) [5,18]. Sulfate taken up from the soil is transported to the shoots and primarily metabolized through reductive assimilation to form cysteine (Cys). Then, methionine and glutathione (GSH) are synthesized from Cys [5,18,19,20]. In *Arabidopsis*, two high-affinity sulfate transporters, SULTR1;1 and SULTR1;2, which are localized to the root epidermis and cortex, are responsible for the initial uptake of sulfate from the soil environment [21,22,23,24,25]. The transcript levels of both *SULTR1;1* and *SULTR1;2* are increased by sulfur deficiency (−S), which increases the sulfate uptake rate [21,22,23,24,25]. The −S-induced the expression of *SULTR1;1* and *SULTR1;2* is controlled by the promoter activities of their 5′-upstream regions [26,27,28,29].

Although SULTR1;1 and SULTR1;2 both contribute to sulfate uptake from the roots and their expression is stimulated by −S, they differ with regard to several features related to transcriptional regulation. When sulfate is sufficiently supplied, the transcript levels of *SULTR1;2* are higher than those of *SULTR1;1* are [24,25]. Under–S conditions, the transcript levels of *SULTR1;2* are still higher than those of *SULTR1;1*, but the rate of induction by −S is less in *SULTR1;2* than that in *SULTR1;1* [23,24,25,30,31]. The sulfate uptake rate and the levels of Cys and GSH in *SULTR1;1* and *SULTR1;2* knockout lines indicate that SULTR1;2 is the main contributor to sulfate uptake under sulfur sufficiency (+S) and −S conditions [23,24,25,30]. In addition, the tissue specificity of *SULTR1;1* and *SULTR1;2* expression in root surface cell layers is not identical: *SULTR1;1* is mainly expressed in the epidermis, including root hairs, whereas *SULTR1;2* is mainly expressed in the cortex [21,24,26].

The genetic control of root hair development has been extensively studied in *Arabidopsis* [6,7,9,32,33,34,35]. Root epidermal cells are separated into root hair and non-hair cells in *Arabidopsis*. Many transcription factors function in the early events of epidermal cell differentiation. In non-hair cells, a core transcriptional complex consisting of TRANSPARENT TESTA GLABRA1 (TTG1, WD40 repeat protein, [36]), GLABRA3 (GL3, bHLH transcription factor, [37]), ENHANCER of GLABRA3 (EGL3, bHLH transcription factor, [38]), and WERWOLF (WER, R2R3 MYB transcription factor, [39]) promotes the expression of the transcription factor GLABRA2 (GL2, homeobox-leucine zipper protein, [40]) which negatively regulates root hair cell-specific genes and positively regulates non-hair cell specific genes to determine non-hair cell fate. In root hair cells, a different protein complex, comprising R3 MYB transcription factors CAPRICE (CPC, [41]), TRIPTYCHON (TRY, [42]), and ENHANCER of TRY and CPC1 (ETC1, [43,44]), inhibits the association of the WER protein with the transcriptional complex described above, thereby repressing GL2 [45,46]. Therefore, given their functions in root hair development, mutation in WER results in an increase in the number of root hairs and mutation in CPC results in a decrease in the number of root hairs.

In addition to these transcription factors, which function as cell fate determinants, many other factors involved in root hair growth have been identified [35]. Among them, a basic Helix-Loop-Helix (bHLH) transcription factor, ROOT HAIR DEFECTIVE 6 (RHD6), plays key roles in determining root hair identity under the control of CPC and GL2 [35,47,48,49]. RHD6 promotes the expression of other bHLH transcription factors including RHD6-LIKE4 (RSL4), which positively regulates root hair growth [50]. The expression of RSL4 is also induced by ethylene through direct induction by the transcription factors ETHYLENE INSENSITIVE 3 (EIN3) and its homolog EIN3-LIKE1 (EIL1) [17]. EIN3 and EIL1 physically interact with RHD6 to synergistically induce *RSL4* expression [17]. As phosphate starvation increases the level of EIN3 protein [16], a phosphate starvation-responsive increase in the number of root hairs would also be mediated by RSL4 and the MYB transcription factor PHOSPHATE STARVATION RESPONSE 1 (PHR1) which plays a central role in phosphate starvation responses [51,52].

In the present study, we investigated the role of root hairs in sulfate uptake using *werwolf* (*wer*) and *caprice* (*cpc*) *Arabidopsis* mutants, as CPC and WER control epidermal cell fate upstream of the signal transduction system of root hair development [6,7,9,32,33,34,35]. Our results indicated that the fate of root epidermal cells influenced the expression of *SULTR1;1* and *SULTR1;2*. However, the –S-induced expression of *SULTR1;1* and *SULTR1;2,* and sulfate uptake were independent of root epidermal cell development. Overall, the sulfate uptake rate was likely influenced by the total root biomass rather than the number of root hairs under both +S and –S conditions.

## 2. Results

### 2.1. SULTR1;1 and SULTR1;2 Were Differentially Expressed in the Root Epidermal Cells under −S

The difference in tissue-specific expression between *SULTR1;1* and *SULTR1;2* in the root surface was investigated by assessing green fluorescent protein (GFP) fluorescence in the root elongation zones of *P_SULTR1;1_-GFP* [26] and *P_SULTR1;2_-GFP* [27] plants. When the plants were grown under +S sulfur sufficient (S1500) conditions, GFP fluorescence was not detected in either plant line (data not shown). Under −S sulfur deficient (S15) conditions, GFP fluorescence was clearly visible in root elongation zones (Figure 1). In *P_SULTR1;1_-GFP* plants, GFP fluorescence was mainly detected in root hair cells (Figure 1a–c), whereas it was detected in non-hair cells in *P_SULTR1;2_-GFP* plants (Figure 1g–i). Interestingly, GFP fluorescence was detected in a striped pattern in the root hair cells and non-hair cells in *P_SULTR1;1_-GFP* and *P_SULTR1;2_-GFP* plants, respectively, suggesting that they were differentially expressed in these cell lines and played different roles in sulfate uptake from the root surface.

### 2.2. Primary Root Length and Numbers of Root Hairs Compensated for Each Other in wer and cpc

To assess the relationship between sulfate uptake and root hair development, root phenotypes of *wer*, *cpc* and wild-type Columbia plants (WT) under S1500 and S15 conditions were observed (Figure 2). The number of root hairs 5 mm from the root tip were counted in plants grown under S1500 and S15 conditions (Figure 2a–c). The number of root hairs was significantly higher under S15 conditions than under S1500 conditions (Figure 2c), suggesting that plants increased the number of root hairs for more efficient sulfate uptake under –S. Under both sulfur conditions, *wer* had more root hairs than the WT, and *cpc* tended to have less root hairs than the WT, as reported previously [39,41] (Figure 2c).

Primary root lengths were not influenced by sulfate concentrations (Figure 2d). Interestingly, primary root lengths were influenced in the opposite direction as the number of root hairs in the two mutants as compared to the WT. That is, primary root lengths tended to be shorter in *wer* and longer in *cpc* than they were in WT under both the sulfur conditions. Significant negative correlations between the number of root hairs and primary root lengths were detected using Sperman’s Rank correlation coefficient (R = 0.047, correlation constant = −0.89). These phenotypes suggested that plants adjusted their root biomass by balancing the inverse relationship between primary root lengths and root hair densities.

Both shoot and root fresh weights were decreased under –S conditions (Figure 2e). In *wer*, shoot fresh weights under S1500 and S15 conditions tended to be less than those in the WT (Figure 2e). When plants were grown under the S15 condition, the root fresh weights of *wer* and *cpc* tended to be less than those in the WT (Figure 2e).

### 2.3. Regulatory Gene Expression in Root Hair Development Was Consistent with Root Hair Numbers under +S but Not –S Conditions

The number of root hairs was increased under –S conditions and, therefore, we analyzed the expression of transcription factors *CPC, WER, GL2, RHD6*, and *RSL4*, which regulate root hair development (Figure 3).

The expression of *CPC* in *wer* was higher under the S1500 condition but lower under the S15 condition than that in the WT, whereas the expression of *WER* was not influenced in *cpc* (Figure 3a). The expression of *GL2* was higher in *cpc* under both conditions than that in the WT.

The transcript levels of root hair-specific transcription factors, *RHD6* and *RSL4*, were higher in *wer* under the S1500 condition than they were in WT but were similar among the genotypes under S15 (Figure 3b). Their transcript levels tended to be lower in *cpc* under both conditions than they were in the WT, which was likely due to the increased expression of *GL2* in *cpc* as GL2 suppressed the expression of *RHD6* [49]. With regard to the expression of *RHD6* and *RSL4,* an interaction was found between sulfur conditions and genotypes using a two-way ANOVA, suggesting their expressions were differentially influenced by the disruption of WER or the number of root hairs under +S and −S.

### 2.4. Sulfate Uptake Was Constant among WT, wer, and cpc

We analyzed the transcript levels of *SULTR1;1* and *SULTR1;2* in the roots and the sulfate uptake activity in WT, *wer,* and *cpc* plants grown under S1500 and S15 conditions (Figure 4). Sulfur deficiency increased the transcript levels of *SULTR1;1* and *SULTR1;2* in all plant lines (Figure 4a). Under the S1500 condition, transcript levels of *SULTR1;2* were similar between the mutants and WT, whereas those of *SULTR1;1* were higher in *wer* and lower in *cpc* than they were in WT (Figure 3a), which is consistent with previous reports [53,54]. These results indicated that the expression of *SULTR1;1* was correlated with the number of root hairs when there was sufficient sulfate for plant growth (Figure 2b,c and Figure 4a).

However, under –S, transcript levels of *SULTR1;1* were lower in *wer* than they were in *cpc,* indicating a negative correlation with the number of root hairs (Figure 2b,c and Figure 4a). Supporting this observation, interactive effects between sulfur conditions and genotypes on the expression of *SULTR1;1* and *SULTR1;2* were detected using a two-way ANOVA, which indicates that the effects of the number of root hairs on their expression would be switched from positive to negative with environmental changes from +S to −S.

The sulfate uptake rate was stimulated under the S15 condition, which was consistent with previous reports [23,24,25,27,55,56] (Figure 4b). Despite the differences in *SULTR1;1* expression between the plant lines, sulfate uptake rates were relatively constant between the two lines under both S1500 and S15 conditions (Figure 4b).

The levels of sulfur-containing metabolites, sulfate, Cys, GSH and total sulfur in the precipitate after extraction with 10 mM HCl were decreased under the S15 condition in all plant lines (Figure 5). Similar to the sulfate uptake rate, the content of sulfate, Cys, and GSH in shoots was generally not influenced by mutations in *WER* and *CPC* under both S1500 and S15 conditions, except for slightly higher sulfur levels in the precipitate of *wer* under the S15 condition than in the WT.

The sum of sulfur levels analyzed in this study was converted to sulfur content in shoot per plant (Figure 6). Concentrations of sulfate, Cys, and GSH (Figure 5) were converted to sulfur content, and added to the sulfur content of the precipitate, and then the total was multiplied by the shoot FW per plant (Figure 2e). The sum of sulfur levels was lower in *wer* under the S1500 condition than it was in the WT and was similar among the genotypes under the S15 condition (Figure 6). Interaction between sulfur conditions and genotypes was detected using a two-way ANOVA, suggesting that sulfur levels were differentially controlled by the number of root hairs under +S and −S conditions.

## 3. Discussion

The mechanisms by which cellular differentiation influences the cellular biochemical or physiological functions and, vice versa, how cellular function influences cell fate, have both long been topics of discussion. In the case of sulfate uptake, two high-affinity sulfate transporters, SULTR1;1 and SULTR1;2, show different spatial expression patterns in root epidermal cells [21,24,26] (Figure 1). Specifically, we found that the expression of *SULTR1;1* was induced preferentially in the root-hair cells in a −S condition [57], whereas *SULTR1;2* was preferentially expressed in non-hair cells (Figure 1). Additionally, the expression of *SULTR1;1* was reported to be higher in *wer* and lower in *cpc* than in WT, whereas the expression of *SULTR1;2* was not influenced in the mutants [53,54]. Based on this information, we postulated that a decrease in the number of root hairs would reduce sulfate uptake activity. However, an increase or decrease in the number of root hairs did not influence the sulfate uptake rate in plants under either +S or −S conditions (Figure 4b). These results indicated that the number of root hairs does not contribute to the sulfate uptake rate, at least in the comparison among WT, *wer* and *cpc*.

Although we observed no difference in sulfate uptake between the WT and mutants, the contribution of root hairs to sulfate uptake cannot be completely ruled out because there was a negative correlation between the number of root hairs and primary root length [58] (Figure 2). The negative correlation suggested that there might be a mechanism to maintain a constant root biomass. A similar negative correlation was observed when plants were exposed to phosphate-deficient conditions, ethylene, or both [16,59], suggesting the involvement of general homeostatic mechanisms. The molecular machinery coordinating the negative interaction between root hair development and primary root length would be an interesting subject to investigate as a potential tool to improve the understanding of nutrient acquisition by plants. The S15 condition also induced a slight but significant increase in the number of root hairs (Figure 2c), which could contribute to the increase in sulfate uptake rate under –S conditions. These results suggested that the sulfate uptake rate was influenced by the total root biomass rather than the number of root hairs under both +S and –S conditions.

The expression of both *SULTRs* was correlated with the number of root hairs negatively under the S15 condition, but positively under the S1500 condition (Figure 2c and Figure 4a). A similar tendency was observed for the expression of *RHD6* and *RSL4* (Figure 3b). Under the S1500 condition, the expression levels of *RHD6* and *RSL4* were higher in *wer* and tended to be lower in *cpc* than they were in the WT, which was consistent with the number of root hairs (Figure 2c and Figure 3b). However, under the S15 condition, the expression of *RHD6* and *RSL4* was similar among the plant lines, whereas the number of root hairs varied in the mutants in a manner similar to that under S1500 condition (Figure 2c and Figure 3b). Although it is not clear whether the expression of *RHD6*, *RSL4,* and *SULTRs* was regulated by the same mechanism under –S conditions, these results suggested that another mechanism likely increased root hair development under –S conditions.

Furthermore, phosphate deficiency induces *RSL4* expression, probably by increasing the stability of EIN3 and EIL1, which both bind to cognate binding sites in the promoter of *RSL4* [16,17]. In another root hair-less line, NR23, several events are induced by phosphate deficiency, including increased the expression of phosphate transporter genes and secretion of acid phosphatases and organic acids, compared to the WT [8]. This observation also supports the differing contribution of root hairs to the uptake of phosphate and sulfate, which varies with the chemical forms of nutrients in the soil [3,4].

Although there were negative or positive correlations between the expression of *SULTR*s and the number of root hairs under both +S and −S conditions, the sulfate uptake rate did not fluctuate among the plant lines (Figure 4). This finding suggests that the transcript levels of *SULTR1;1* and *SULTR1;2* were not the only determinants of sulfate uptake rate. Further studies of the relationship between root architecture and sulfate uptake may shed light on the as yet unknown determinants of this process.

## 4. Materials and Methods

### 4.1. Plant Materials and Growth Conditions

*Arabidopsis thaliana* plants, ecotype “Columbia” (Col-0), were used as the wild-type (WT) while *cpc* [41] (*cpc-1*) and *wer* [39] (*wer-1*) mutants were obtained from the Arabidopsis Biological Resource Center (ABRC). Plants were grown at 22 °C under continuous light (40 µmol m^−1^ s^−1^) conditions on mineral nutrient media containing 1% sucrose [60,61]. For the preparation of the agar medium, agar was washed twice with 1 L de-ionized water and vacuum filtered to remove the sulfate. Sulfur sufficient (S1500) agar medium was supplemented with 1500 µM MgSO_4_. Sulfur-deficient (S15) agar medium was supplemented with 15 µM MgSO_4_ and Mg concentration was adjusted to 1500 µM by adding MgCl_2_. After the indicated shown in each figure, the shoot and root tissues were harvested separately, rinsed with distilled water, and subsequently subjected to various analyses.

### 4.2. Observation of GFP Fluorescence

The tissue-specific expression of GFP in *P_SULTR1;1_-GFP* and *P_SULTR1;2_-GFP* transgenic plants was visualized in whole mounts of 7-day-old plants using a fluorescent microscope system (EVOS FL Auto 2 Imaging System) equipped with the EVOS Light Cube, GFP (Ex: 470/22, Em: 525/50) (Thermo Fisher Scientific, USA).

### 4.3. Observation of Root Development

The primary root lengths of plants were analyzed using images captured with a STAGE2000-BG system (AMZ System Science, Japan). The number of root hairs in 5 mm from the root tip was analyzed from the image captured using a CCD camera (WRAYCAM G500, WRAYMER, Japan) connected to a stereoscopic microscope (SW-700TD, WRAYMER). The free software package, ImageJ [62,63] was used for the analysis.

### 4.4. Quantitative Real-Time RT-PCR Analysis

RNA preparation and RT were performed as reported previously [27,54,55]. Real-time PCR was carried out using a SYBR Green Perfect real-time kit (Takara, Japan) and Thermal Cycler Dice real-time system (Takara) using the gene-specific primers for CPC, CPC-F (5′-GGATGTATAAACTCGTTGGCGACAG-3′) and CPC-R (5′-GCCGTGTTTCATAAGCCAATATCTC-3′) [64]; for WER, WER-F (5′-TGGTAATAGGTATAACTTCATTTGC-3′) and WER-R (5′-TTTGATTCCGAGTTTCTTACTAAGGATG-3′); for GL2, GL2-F (5′-TCGGATCACTGAGACCACAA-3′) and GL2-R (5′-GTGTATCCCGGAACCAGTGT-3′) [64]; for RHD6, RHD6-real-F (5′-TGATTTGGTGACAATGCTTGA-3′) and RHD6-real-R (5′-GGAGAGAATGGCATCAATGG-3′) [49]; for RSL4, RSL4_q-PCR f new (5′-AACCTTGTGCCAAACGGGAC-3′) and RSL4_q-PCR r new (5′-CCAGGCCGTTGTAAGCCAAT-3′) [17]; and, for *SULTR1;2*, SULTR1;2-1854F (5′-GGATCCAGAGATGGCTACATGA-3′) and SULTR1;2-1956R (5′-TCGATGTCCGTAACAGGTGAC-3′) [27]; for *SULTR1;1*, SULTR1;1-625F (5′-GCCATCACAATCGCTCTCCAA-3′) and SULTR1;1-750R (5′-TTGCCAATTCCACCCATGC-3′) [30]; and for ubiquitin, UBQ2-144F (5′-CCAAGATCCAGGACAAAGAAGGA-3′) and UBQ2-372R (5′-TGGAGACGAGCATAACACTTGC-3′) [30]. The relative mRNA levels were calculated using ubiquitin2 as an internal standard.

### 4.5. Sulfate Uptake Assay

Plants were vertically grown for 10 days on S1500 and S15 agar media. The roots were submerged in S1500 medium containing 15 µM [^35^S] sodium sulfate (American Radiolabeled Chemicals, USA) and incubated for 1 h. Washing and measurement were carried out as described previously [25,27,54,55,65,66].

### 4.6. Measurement of Sulfate, Cysteine and Glutathione, and Total Sulfur Levels

Plant tissues were frozen in liquid nitrogen and homogenized in 5 volumes of 10 mM HCl using a Tissue Lyser (Retsch, Germany). After homogenization, the cell debris was removed by centrifugation, and the supernatant was subsequently analyzed.

For sulfate measurement, the extracts were diluted 100-fold with extra pure water and analyzed using ion chromatography (IC-2001, TOSOH, Japan). Using serial 30-µL injections, the diluted extracts were separated at 40 °C using a TSK SuperIC-AZ column (TOSOH) at a flow rate of 0.8 mL min^−1^ with an eluent containing 1.9 mM NaHCO_3_ and 3.2 mM Na_2_CO_3_. Anion mixture standard solution 1 (Wako Pure Chemicals, Japan) was used as a standard.

Cys and GSH contents were determined using monobromobimane (Invitrogen, USA) labeling of the thiols after reduction of the plant extracts with dithiothreitol (DTT). The labeled products were then separated using HPLC (JASCO, Japan) using the TSKgel ODS-120T column (150 × 4.6 mm, TOSOH) and detected using a fluorescence detector, FP-920 (JASCO), monitoring for fluorescence of thiol-bimane adducts at 482 nm under excitation at 390 nm. Cys and GSH (Nacalai Tesque, Japan) were used as standards.

The total sulfur content was determined using inductively coupled plasma-mass spectroscopy (ICP-MS; Agilent7700, Agilent Technologies, USA). The precipitates obtained from the extraction described above were digested in 200 µL HNO_3_ at 95 °C for 30 min and then 115 °C for 90 min. After cooling to room temperature, the digested samples were diluted to 1 mL with extra pure water, filtered using 0.45 µm filters (DISMIC-03CP, ADVANTEC, Japan). The filtered samples were diluted 10 times with a solution consisting of 0.1 M HNO_3_ and 10 µg L^−1^ gallium (KANTO CHEMICAL, Japan) as an internal standard, before being subjected to ICP-MS. Quantification was performed using the standard curve obtained via serial dilutions of the sulfur standard solution (KANTO CHEMICAL).

### 4.7. Statistical Analysis

Two-way ANOVA was used to detect the effects of sulfur conditions, genotypes, and their interactions (Figure 2, Figure 3, Figure 4, Figure 5 and Figure 6). Significant differences between S1500 and S15 conditions detected using a two-way ANOVA are indicated with asterisks (*P* < 0.05). Furthermore, where an interaction was detected between sulfur conditions and genotypes, the Tukey–Kramer test was applied to all experimental conditions (Figure 3 and Figure 4a). In addition, where interaction was not detected between the values, the Tukey–Kramer test was used to analyze all genotypes grown under the same growth condition. Significant differences detected by the Tukey–Kramer test were shown with different letters (*P* < 0.05). Statcel4 software (OMS Publishing Inc., Tokyo, Japan) was used for all statistical analysis with the Microsoft Excel program.

## Figures and Tables

**Figure 1 plants-08-00106-f001:**
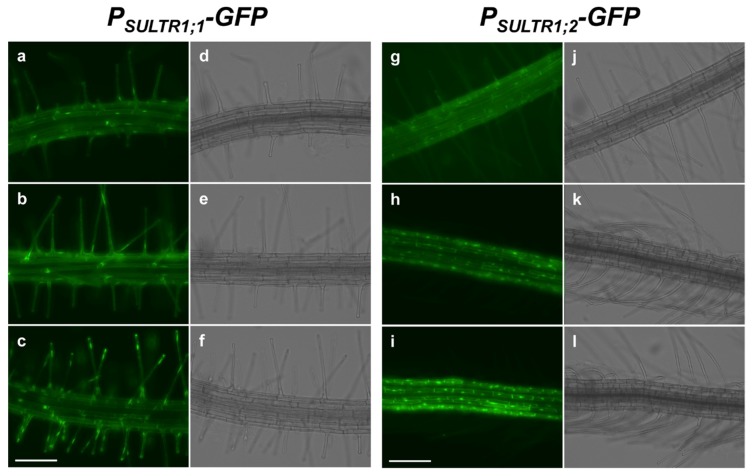
Expression of green fluorescent protein (GFP) in the root elongation zones of *P_SULTR1;1_-GFP* plants (**a**–**f**, [26]) and *P_SULTR1;2_-GFP* plants (**g**–**l**, [27]). Three *T_2_* generation lines carrying each construct were grown for 7 d on agar media supplied with 15 μM sulfate and observed using fluorescent microscopy (EVOS FL Auto 2). GFP images (**a**–**c**,**g**–**i**) and bright field images (**d**–**f**,**j**–**l**) are presented. Scale bars, 100 µm.

**Figure 2 plants-08-00106-f002:**
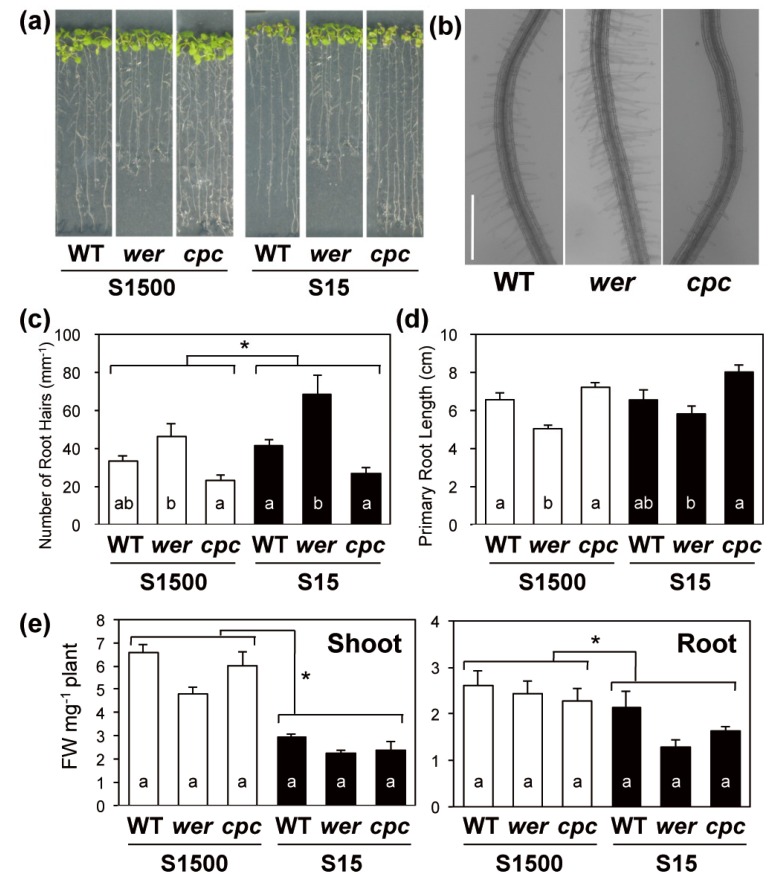
Plant growth and root development of wild-type (WT), and *werwolf* (*wer*), and *caprice* (*cpc*) mutants grown under sulfur-sufficient and -deficient conditions. Plants were vertically grown for 10 d on agar media supplied with 1500 μM sulfate (sulfur sufficient, S1500, white bar) or 15 μM sulfate (sulfur-deficient, S15, black bar). (**a**) Images of plants on their media. (**b**) Typical root image of plants grown under the S15 condition. Scale bar, 500 µm. (**c**) Number of root hairs in root elongation zone. Number of root hairs 5 mm from root tip was counted. (**d**) Primary root lengths. (**e**) Shoot and root fresh weights (FW) per plant. Values and error bars indicate means ± SEM [n = 5 in (**c**), n = 8 to 12 in (**d**), and n = 3 in (**e**)]. Two-way ANOVA was used to detect the effects of sulfur conditions, genotypes, and their interactions. Asterisks indicate significant differences between the S1500 and S15 conditions (two-way analysis of vatiance [ANOVA], * *P* < 0.05). As all comparisons in (**c**–**e**) did not detect interaction, the Tukey–Kramer test was used to analyze significant differences among mutants and the WT under the same growth condition. Different letters indicate significant differences (*P* < 0.05).

**Figure 3 plants-08-00106-f003:**
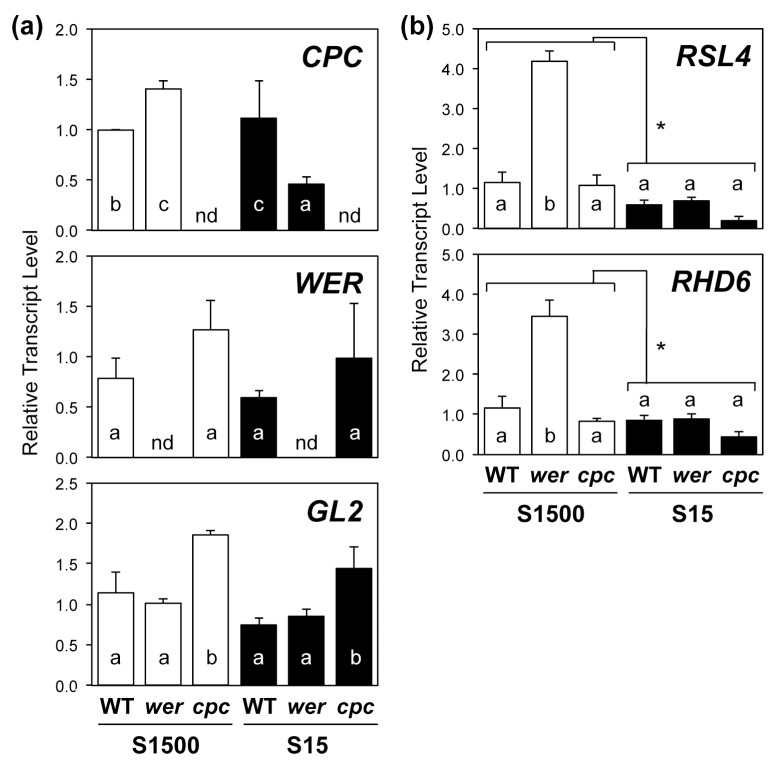
Expression of transcription factors regulating root hair development in wild-type (WT), *werwolf* (*wer)*, and *caprice* (*cpc*) mutants grown under S1500 and S15 conditions. (**a**) Relative transcript levels of *CPC, WER,* and *GL2,* and (**b**) those of *RHD6*, and *RSL4,* in root tissues were determined using quantitative reverse transcription-polymerase chain reaction (qRT-PCR). Plants were vertically grown for 10 d on S1500 (white bars) and S15 (black bars) agar media. Values and error bars indicate means ± SEM (n = 3). Two-way analysis of variance (ANOVA) was used to detect the effects of sulfur conditions, genotypes, and their interactions. In (**a**), all comparisons did not detect interactions, so the Tukey–Kramer test was applied to the three genotypes under the same growth conditions. In (**b**), interactions between sulfur conditions and genotypes were detected, so the Tukey–Kramer test was applied to all six experimental conditions. Different letters indicate significant differences (*P* < 0.05).

**Figure 4 plants-08-00106-f004:**
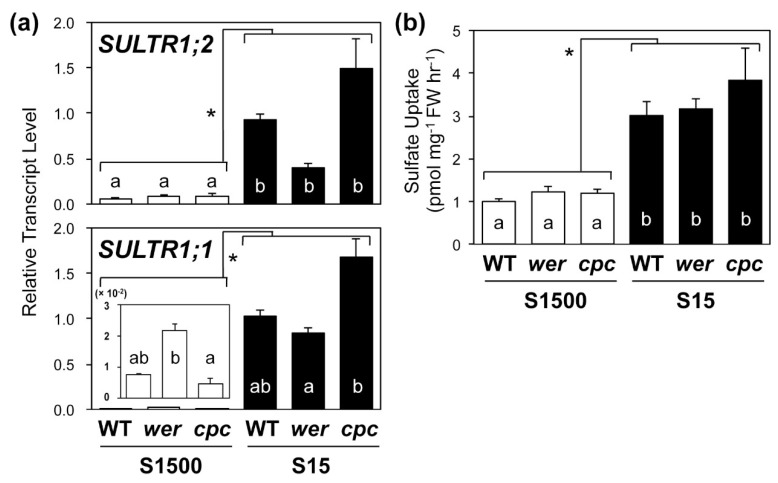
Transcript levels of *SULTR1;1* and *SULTR1;2,* and sulfate uptake activity in wild-type (WT), *werwolf* (*wer*), and *caprice* (*cpc*) mutants grown under S1500 and S15 conditions. (**a**) Relative transcript levels of *SULTR1;1* and *SULTR1;2* in root tissues determined using quantitative reverse transcription-polymerase chain reaction (qRT-PCR). (**b**) Sulfate uptake activity of plants. Absolute values of [^35^S] sulfate uptake rates are presented. Plants were vertically grown for 10 d on S1500 (white bars) and S15 (black bars) agar media. Values and error bars indicate means ± SEM [n = 3 in (**a**), n = 6 in (**b**)]. Two-way analysis of variance (ANOVA) was used to detect the effects of sulfur conditions, genotypes, and their interactions. Asterisks indicate significant differences between S1500 and S15 conditions (two-way ANOVA; * *P* < 0.05). In (**a**), interactions between sulfur conditions and genotypes were detected and, so the Tukey–Kramer test was applied to all six experimental conditions. In (**b**), no interaction was detected and, so the Tukey–Kramer test was applied to the three genotypes grown under the same condition. Different letters indicate significant differences (*P* < 0.05).

**Figure 5 plants-08-00106-f005:**
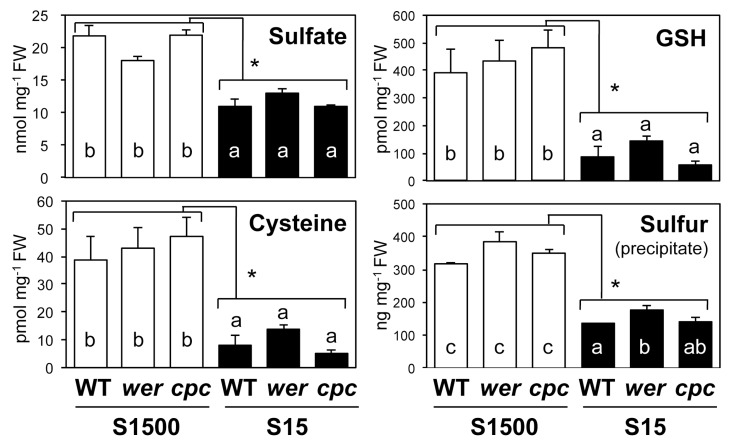
Accumulation of sulfate, cysteine (Cys), glutathione (GSH), and sulfur in shoots of wild-type (WT), *werwolf* (*wer*), and *caprice* (*cpc*) mutants grown under S1500 and S15 conditions. Plants were vertically grown for 10 d on S1500 (white bars) and S15 (black bars) agar media. Sulfate content in shoot tissues was determined using ion chromatography. Cys and GSH content of shoot tissues was analyzed using a high-performance liquid chromatography (HPLC)-fluorescent detection system, after labeling thiol bases with monobromobimane. Total sulfur content in precipitates was analyzed using inductively coupled plasma-mass spectroscopy (ICP-MS) after nitric acid digestion. Values and error bars indicate means ± SEM (n = 3). Two-way analysis of variance (ANOVA) was used to detect effects of sulfur conditions, genotypes, and their interactions. Asterisks indicate significant differences between S1500 and S15 conditions (two-way ANOVA; * *P* < 0.05). As all comparisons did not detect interactions, the Tukey–Kramer test was applied to detect significant differences among genotypes grown under similar conditions. Different letters indicate significant differences (*P* < 0.05).

**Figure 6 plants-08-00106-f006:**
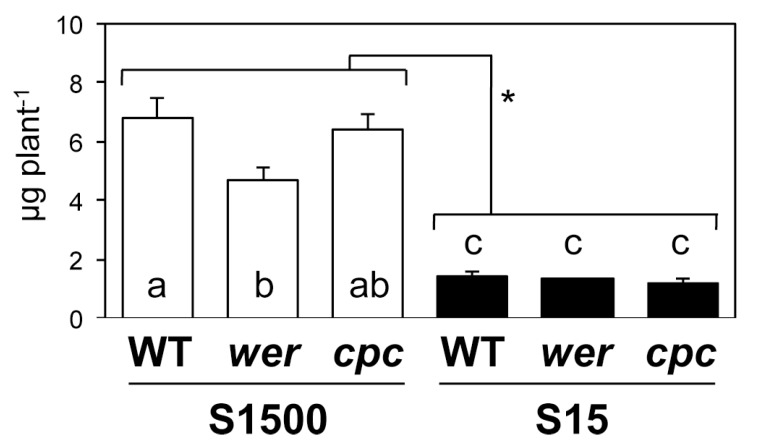
Sum of sulfur levels in shoots of WT, *wer*, and *cpc* mutants grown under S1500 and S15 conditions. Plants were vertically grown for 10 d on S1500 (white bars) and S15 (black bars) agar media. Sulfate, Cys, GSH, and sulfur contents in precipitate of shoot tissues were calculated to µg sulfur per plant. The values and error bars indicate means ± SEM (n = 3). As two-way ANOVA detected the interaction between sulfur conditions and genotypes, the Tukey–Kramer test was applied to all six experimental conditions. Different letters mean significant differences (*P* < 0.05). Asterisks indicate significant differences between S1500 and S15 conditions (2-way ANOVA; * *P* < 0.05).

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
