# Peer review of "Contribution of Root Hair Development to Sulfate Uptake in *Arabidopsis"

_plants, 2019, doi:10.3390/plants8040106_

Round 1
Reviewer 1 Report
This is an elegant experiment that tests the role of root hairs in mediating sulphur uptake when Arabidopsis plants are grown on sulphur-sufficient and sulphur-deficient medium, by using mutants with more or less root hairs. Gene expression of sulphate transporters is measured, along with root morphology. Sulphur uptake was independent of root hair morphology, although some morphological traits changed in the mutants. A few comments
Please justify the selection of the specific mutants. There are root hairless mutants of Arabidopsis, which may have provided a better contrast with the WT
Please calculate total S content of the different plants (ie. Total S concentration of roots and shoots, multiplied by root and shoot biomass)
Please improve the statistical analysis as discussed below, and use this to more clearly express the Results section
Line 18 Sentence should be subdivided – start with “Sulfate uptake rates were…”, and then indicate whether S deficiency enhanced S uptake rate
Then new sentence “Under S-, expression of….”
Line 22 Again, please subdivide the sentence – there are 3 important pieces of information therein
Line 204 “negative correlation” – perhaps a correlation table would be useful ?
Line 230 “filtered” not “filtrated”
Line 232 How can the authors be sure that morphological responses were due to sulphate deficiency and not chloride toxicity ? A dose-response of root growth comparing K salts and Mg salts of chloride & sulphate would seem prudent
The statistical analysis in Figure 2c-e (and all other histograms) is primitive, use 2 way ANOVA to distinguish effects of S supply,, genotype and their interaction. Each mutant should be compared with the WT to determine whether it responds different to medium S status (as indicated by a significant genotype x S concentration interaction). Then using 1 way ANOVA and a post-hoc test, add letters of mean discrimination above the bars in Figure 2, to clearly indicate which genotype / S status combinations significantly differ from each other
I am surprised the S analyte data are expressed per unit fresh weight – presumably leaf water content was the same between treatments / genotypes, but better to express per unit dry weight
Author Response
Responses to Reviewer 1 comments
> Please justify the selection of the specific mutants. There are root hairless mutants of Arabidopsis, which may have provided a better contrast with the WT.
Thank you for the critical comment. As CPC and WER are located on the upstream of transcription factors for epidermal cell differentiation in Arbidopsis, these two transcription factors are generally recognized for their importance to determine the epidermal cell fate. That is the reason why we selected cpc and wer to analyze the effects on sulfate uptake. According to the comment, we have added the explanation in L88-90.
> Please calculate total S content of the different plants (ie. Total S concentration of roots and shoots, multiplied by root and shoot biomass).
Thank you for the helpful comment. According to the comment, we have calculated total S content in all tested conditions from the metabolite levels analyzed in this study. The results have been added as Figure 6 and described in L226-233.
> Please improve the statistical analysis as discussed below, and use this to more clearly express the Results section.
We appreciate the critical comment. According to the comment, we have changed the statistical analysis in Figures 2 to 6, revised the related description in Results, and added the section “4.7 Statistical Analysis” to Materials and Methods.
> Line 18 Sentence should be subdivided – start with “Sulfate uptake rates were…”, and then indicate whether S deficiency enhanced S uptake rate
Then new sentence “Under S-, expression of….”
According to the comments, we have subdivided the sentence into two sentences in L17-20.
> Line 22 Again, please subdivide the sentence – there are 3 important pieces of information therein
According to the comments, we have subdivided the sentence by using the numbers in L23-24.
> Line 204 “negative correlation” – perhaps a correlation table would be useful ?
Thank you for the helpful comment. In this part, we described the negative correlation between the number of root hairs and primary root length, so we have analyzed the correlation with Sperman’s Rank correlation coefficient and described the result in L140-142.
> Line 230 “filtered” not “filtrated”
“filtered” seems fine. Upon the revision, we have asked professional editing to correct grammatical errors (Editage Inc). Please find the attached certificate of language editing.
> Line 232 How can the authors be sure that morphological responses were due to sulphate deficiency and not chloride toxicity ? A dose-response of root growth comparing K salts and Mg salts of chloride & sulphate would seem prudent
Thank you for the thoughtful comments. Because it is the commonly accepted conditions to analyze the effects of sulfur deficiency, this time we didn’t add the analysis with K salts. We can exclude the possibility that we had observed toxic effects of chloride because there was no visible effect on plant growth when the same level of MgCl2 (1.5 mM) was added to the media supplemented with 1.5 mM MgSO4 [Zhang et al. (2014) Plant J. 77: 185-197]. Also, the toxic effects of NaCl can be observed with the concentration higher than 20 mM [Mol. Cell. Biol. (2007) 27: 5214-5224, Shavrukov (2012) J. Exp. Bot. 64: 119-127, Wu (2018) Crop J. 6: 215-225], which is much higher than the chloride concentration in our media (1.5 mM).
> The statistical analysis in Figure 2c-e (and all other histograms) is primitive, use 2 way ANOVA to distinguish effects of S supply,, genotype and their interaction. Each mutant should be compared with the WT to determine whether it responds different to medium S status (as indicated by a significant genotype x S concentration interaction). Then using 1 way ANOVA and a post-hoc test, add letters of mean discrimination above the bars in Figure 2, to clearly indicate which genotype / S status combinations significantly differ from each other
Thank you for the critical comments. According to the comments, we thoroughly revised the statistical analysis with two-way ANOVA and the following post-hoc test depending on the interaction between sulfur conditions and genotypes. Figures and the related descriptions in Figure legends, Results, and Discussion have been revised and the section “4.7 Statistical Analysis” has added to Materials and Methods.
> I am surprised the S analyte data are expressed per unit fresh weight – presumably leaf water content was the same between treatments / genotypes, but better to express per unit dry weight.
We agree with the comment. However, as we analyze the metabolites and sulfur contents with the plant fresh weight from 10 mg to 50 mg, it is difficult to weigh the exact dry weight. When we had compared the dry weight between sulfur sufficient and deficient conditions with much higher fresh weight of the plants, water content was similar between the conditions, so we can apply the dry weight calculated by the former analysis. However, as we did not weigh those for the mutants, so we think it is more accurate to show the concentrations per fresh weight.
Reviewer 2 Report
REVIEW
The authors present an interesting work on two high-affinity sulfate transporters SULTR1;1 and SULTR1;2 and provide evidence that sulfate uptake is influenced by root surface area rather than root hair numbers. To answer these questions, the authors employ confocal imaging, RT-qPCR, physiological assays and metabolomics.
The manuscript demonstrates that SULTR1;1 expression overlaps with root hairs, whereas SULTR1;2 is expressed in the cortex. Based on these observations, the authors reveal that wer and cpc mutants, either with increased or reduced root hair numbers, respectively, still express SULTR1;1 and SULTR1;2 under low sulfur concentration.
Taken together, the manuscript is written in a concise way, the figures are presented in a logical order and the experiments are nicely conceived. I am quite enthusiastic about the manuscript. I have raise some minor points and would appreciate if authors could consider them prior to manuscript acceptance.
Minor points:
1. page 2, line 87f.:I would appreciate if the authors could show confocal images of plants expressing GFP from SULTR1;1 and SULTR1;2 promoters under sulfur sufficiency (+S). Although the authors have mentioned that a GFP fluorescence signal was not detected, it should be included in the manuscript and serve as an internal control.
2. page 5, line 148ff.:The authors should provide a more concise statement on ‘the number of root hairs on the expression of SULTR1;1 and SULTR1;2 would be different between +S and -S’. The paragraph’s summary is too brief and not fully conclusive. Requires rephrasing.
3. page 8, line 199f. and 204ff.:The authors discuss that root hair numbers do not influence sulfate uptake, however rather suggest that a homeostatic mechanism with characteristics on an inversely correlated mutual interdependency between primary root length and root surface may exist and determine sulfate uptake rates. Under this hypothesised premise, the authors should measure diameters of primary roots with a particular focus on elongation zone in wildtype, werand cpcunder +S and –S conditions.
4. It has been shown that phosphate starvation can activate RSL4 expression, a crucial regulator for root hair development, most probably through an increased stabilisation of the ethylene responsive transcription factors EIN3/EIL1 which both bind to cognate binding sites within the promoter of RSL4 (Song et al., 2016; Feng et al., 2017). Along the same line, the authors should test whether sulfur might have a transcriptional influence on RSL4 as well as on WER and CPC which would help to understand whether a negative feedback loop might exist and accounts for the magnitude of responsiveness on the developmental process of root hair formation.
5. GL2 constitutes a primary target gene of WER and CPC which in turn repress the expression of root hair responsive genes (e.g. RSL1, RSL2, LRL1and LRL2) (Lin et al., 2015). The observation that wer and cpc mutants display different degree of root hair formation under –S might indicate that GL2 expression could be regulated by sulfur. The authors should test GL2 transcript abundance under –S and +S since GL2, a negative regulator of root hair development, overlaps with its expression pattern with SULTR1;1.
References:
Feng et al., 2017, PNAS, vol.114(52):13834
Lin et al., 2015, The Plant Cell, vol.27:2894
Song et al., 2016, PLoS Genetics, vol.12(7)e1006194
Author Response
Responses to Reviewer 2 comments> 1. page 2, line 87f.: I would appreciate if the authors could show confocal images of plants expressing GFP from SULTR1;1 and SULTR1;2 promoters under sulfur sufficiency (+S). Although the authors have mentioned that a GFP fluorescence signal was not detected, it should be included in the manuscript and serve as an internal control.
Thank you for the kind advises. Although we completely agree with the comment, we did not take the images under sulfur sufficient conditions then as GFP fluorescence was not detected with the conditions. In the case that it is essential, we could do the experiment again, but this time we think the data is supportive but not essential to lead out the conclusion, so we have not taken the images.
> 2. page 5, line 148ff.: The authors should provide a more concise statement on ‘the number of root hairs on the expression of SULTR1;1 and SULTR1;2 would be different between +S and -S’. The paragraph’s summary is too brief and not fully conclusive. Requires rephrasing.
Thank you for the helpful comments. According to the comment, we have rephrased the sentences in L196-201.
> 3. page 8, line 199f. and 204ff.: The authors discuss that root hair numbers do not influence sulfate uptake, however rather suggest that a homeostatic mechanism with characteristics on an inversely correlated mutual interdependency between primary root length and root surface may exist and determine sulfate uptake rates. Under this hypothesised premise, the authors should measure diameters of primary roots with a particular focus on elongation zone in wildtype, werand cpcunder +S and –S conditions.
Thank you for the helpful comments. We have considered this issue, and thought it is not accurate to calculate the root surface area by measuring the diameter of primary roots with a particular focus and primary root length. So we have revised the description “root surface area” to “root biomass” throughout the text. I hope this revision can meet for the comment.
> 4. It has been shown that phosphate starvation can activate RSL4 expression, a crucial regulator for root hair development, most probably through an increased stabilisation of the ethylene responsive transcription factors EIN3/EIL1 which both bind to cognate binding sites within the promoter of RSL4 (Song et al., 2016; Feng et al., 2017). Along the same line, the authors should test whether sulfur might have a transcriptional influence on RSL4 as well as on WER and CPC which would help to understand whether a negative feedback loop might exist and accounts for the magnitude of responsiveness on the developmental process of root hair formation.
Thank you for the thoughtful advice. According to the comment, we have analyzed the expression of some transcription factors regulating root hair development, CPC, WER, GL2, RHD6, and RSL4, and added the data in Figure 3. As the results, the expression of CPC was differentially influenced in wer by sulfur conditions compared to that in WT. Also, the expression of RHD6 and RSL4 were differentially influenced in wer between sulfur sufficient and deficient conditions. Although we could not figure out the regulatory mechanisms to increase the number of root hairs under –S and also the meaning of the expression levels of these genes, these results provide new information about the relationship between root hair development and nutrient acquisition. We have added the related descriptions throughout the manuscript.
> 5. GL2 constitutes a primary target gene of WER and CPC which in turn repress the expression of root hair responsive genes (e.g. RSL1, RSL2, LRL1and LRL2) (Lin et al., 2015). The observation that wer and cpc mutants display different degree of root hair formation under –S might indicate that GL2 expression could be regulated by sulfur. The authors should test GL2 transcript abundance under –S and +S since GL2, a negative regulator of root hair development, overlaps with its expression pattern with SULTR1;1.
Thank you for this advice. As described in 4., expression of GL2 was analyzed in Figure 3. GL2 expression was higher in cpc regardless to the sulfur conditions, but was not significantly influenced by sulfur conditions. Also, the expression of GL2 was not quantitatively correlated with that of SULTR1;1, RHD6 and RSL4, suggesting that root hair development was differentially regulated under –S compared to +S. We have added related descriptions in the text.
> References:
Feng et al., 2017, PNAS, vol.114(52):13834
Lin et al., 2015, The Plant Cell, vol.27:2894
Song et al., 2016, PLoS Genetics, vol.12(7)e1006194
These references were certainly informative, thank you. We have added descriptions about these backgrounds in Introduction and Discussion.
Reviewer 3 Report
In this manuscript author described about
Contribution of root hair development to sulfate uptake in Arabidopsis. This paper need extensive english editing. Please resubmit it after English enditing. Also, Results are not sufficient enough.
Did you check root marker genes responsible for root hairs. Did you correlate it with other pathway involved in root hair developement such as ethylene and auxin?
Discussion did not cover all the aspect. Add more information in this part.
Author Response
Responses to Reviewer 3 comments
>In this manuscript author described about Contribution of root hair development to sulfate uptake in Arabidopsis. This paper need extensive english editing. Please resubmit it after English enditing. Also, Results are not sufficient enough.
Thank you for the critical comments. According to the comment, we have asked professional editing to correct grammatical errors (Editage Inc). Please find the attached certificate of language editing.
Also, we have added some results related to the next comments in Results, the related descriptions in Introduction and Discussion, and revised the statistical analysis for Figures 2 to 6. We hope these revisions could fulfill the reviewer’s requirements.
>Did you check root marker genes responsible for root hairs. Did you correlate it with other pathway involved in root hair developement such as ethylene and auxin?
Thank you for the thoughtful comments. We have analyzed the expression of some transcription factors regulating root hair development, CPC, WER, GL2, RHD6, and RSL4, and added the data in Figure 3. As the results, the expression of CPC was differentially influenced in wer by sulfur conditions compared to that in WT. Also, the expression of RHD6 and RSL4 were differentially influenced in wer between sulfur sufficient and deficient conditions. Although we could not figure out the regulatory mechanisms to increase the number of root hairs under –S and also the meaning of the expression levels of these genes, these results provide new information about the relationship between root hair development and nutrient acquisition. We have added the related descriptions throughout the manuscript.
> Discussion did not cover all the aspect. Add more information in this part.
We have added some descriptions in Discussion. We hope the revision would meet for the comment.
Round 2
Reviewer 3 Report
I am convinced with the author reply, and this MS can be accepted in its current format.